# SERPINE1 mRNA Binding Protein 1 Is Associated with Ischemic Stroke Risk: A Comprehensive Molecular–Genetic and Bioinformatics Analysis of *SERBP1* SNPs

**DOI:** 10.3390/ijms24108716

**Published:** 2023-05-13

**Authors:** Irina Shilenok, Ksenia Kobzeva, Tatiana Stetskaya, Maxim Freidin, Maria Soldatova, Alexey Deykin, Vladislav Soldatov, Mikhail Churnosov, Alexey Polonikov, Olga Bushueva

**Affiliations:** 1Laboratory of Genomic Research, Research Institute for Genetic and Molecular Epidemiology, Kursk State Medical University, 305041 Kursk, Russia; 2Division of Neurology, Kursk Emergency Hospital, 305035 Kursk, Russia; 3Laboratory of Statistical Genetics and Bioinformatics, Research Institute for Genetic and Molecular Epidemiology, Kursk State Medical University, 305041 Kursk, Russia; 4Department of Biology, School of Biological and Behavioural Sciences, Queen Mary University of London, London E1 4NS, UK; 5Laboratory of Population Genetics, Research Institute of Medical Genetics, Tomsk National Research Medical Center, Russian Academy of Science, 634050 Tomsk, Russia; 6Laboratory of Genome Editing for Biomedicine and Animal Health, Belgorod State National Research University, 308015 Belgorod, Russia; 7Department of Pharmacology and Clinical Pharmacology, Belgorod State National Research University, 308015 Belgorod, Russia; 8Department of Medical Biological Disciplines, Belgorod State University, 308015 Belgorod, Russia; 9Department of Biology, Medical Genetics and Ecology, Kursk State Medical University, 305041 Kursk, Russia

**Keywords:** ischemic stroke, hero, heat-resistant obscure, chaperones, SERPINE1 mRNA binding protein 1, SERBP1, SNP, activated partial thromboplastin time, body mass index, gene-environmental interaction

## Abstract

The *SERBP1* gene is a well-known regulator of SERPINE1 mRNA stability and progesterone signaling. However, the chaperone-like properties of *SERBP1* have recently been discovered. The present pilot study investigated whether *SERBP1* SNPs are associated with the risk and clinical manifestations of ischemic stroke (IS). DNA samples from 2060 unrelated Russian subjects (869 IS patients and 1191 healthy controls) were genotyped for 5 common SNPs—rs4655707, rs1058074, rs12561767, rs12566098, and rs6702742 *SERBP1*—using probe-based PCR. The association of SNP rs12566098 with an increased risk of IS (risk allele C; *p* = 0.001) was observed regardless of gender or physical activity level and was modified by smoking, fruit and vegetable intake, and body mass index. SNP rs1058074 (risk allele C) was associated with an increased risk of IS exclusively in women (*p* = 0.02), non-smokers (*p* = 0.003), patients with low physical activity (*p* = 0.04), patients with low fruit and vegetable consumption (*p* = 0.04), and BMI ≥25 (*p* = 0.007). SNPs rs1058074 (*p* = 0.04), rs12561767 (*p* = 0.01), rs12566098 (*p* = 0.02), rs6702742 (*p* = 0.036), and rs4655707 (*p* = 0.04) were associated with shortening of activated partial thromboplastin time. Thus, *SERBP1* SNPs represent novel genetic markers of IS. Further studies are required to confirm the relationship between *SERBP1* polymorphism and IS risk.

## 1. Introduction

Ischemic stroke (IS), which accounts for ~80% of the total cases of stroke [1], is the second leading cause of death and the third leading cause of disability worldwide [2]. IS is caused by an interruption in cerebral blood flow, induced by thrombosis or embolism [3]. In this regard, the main factors contributing to IS are atherogenesis, instability of atherosclerotic plaque, and increased platelet aggregation. Accordingly, genetic studies revealed numerous IS-associated polymorphisms among the genes related to lipids turnover, inflammatory response, oxidative stress, and platelets activity [4,5,6,7]. Nevertheless, further genetic studies may help to reveal other molecular contributors, shedding light on the pathogenesis of IS.

Brain damage during acute ischemia is a very multifaceted and complex process. However, among the major diversity of molecular events, the following scenario can be distinguished as key: unbalanced functioning of the mitochondrion and reduced ATP production cause the accumulation of H^+^ and Ca^2+^ ions as well as free radicals. These toxic molecules directly damage cellular structure and provoke pathological responses displayed as excessive glutamate release and a switch to an inflammatory phenotype [8,9]. Finally, reaching the point of no return, the cell launches the “suicide programme” or worse, perishes via necrosis [10,11,12].

Under ischemia as well as under other pathological conditions, the cell utilizes all accessible resources to resist before it dies. One of the most powerful tools to decrease cellular damage is chaperone machinery. Chaperones refer to a functionally related group of proteins that provide proper folding of newly synthesized and unfolded proteins. The biological tasks of chaperones include providing correct protein folding, translocation of proteins across the membranes, helping in the assembly and disassembly of protein complexes, and participating in their degradation [13,14]. Obviously, the tremendously important role of chaperones determines their crucial significance for the course of brain ischemia. Moreover, being widely involved in the regulation of cellular proteostasis, the chaperones are important players in the development of IS [15].

SERPINE1 mRNA binding protein 1 (SERBP1) has first been identified as a protein bound to plasminogen activator inhibitor type 1 (PAI-1) mRNA [16] and which regulates the stability of the transcript. A significant role of PAI-1 in the increased risk of thrombosis and IS has been shown by a lot of studies [17,18,19]. Subsequent studies discovered that SERBP1 is involved in progesterone signaling via interactions with progesterone receptor membrane component 1 (PGRMC1) [20]. 

In 2020, Tsuboyama K et al. revealed SERBP1 among other small heat-resistant proteins in drosophila cell culture lysate that display chaperone-like activity even after being boiled [21]. As well as the other members, SERBP1 was named “heat-resistant obscure”, or, acronymically, “Hero”. The authors have also found mammalian orthologs of all the Hero-proteins and confirmed their chaperone-like properties. Our previous study has already shown an association of one of the “Hero” genes with the risk of IS [22].

Summarizing the biological roles of *SERBP1* may suggest its probable link with IS. Indeed, the involvement of *SERBP1* in the regulation of PAI-1, one of the main drivers of atherothrombosis, as well as the recently discovered “chaperone” properties of *SERBP1*, make this gene very attractive in determining the factors leading to the risk of IS. As far as such studies have not been previously carried out, we aimed to analyze the associations of functionally significant *SERBP1* genetic variants with the risk and clinical manifestations of IS.

## 2. Results

### 2.1. Associations of SERBP1 SNPs with the Risk of Ischemic Stroke

The genotype frequencies of rs4655707, rs1058074, rs12561767, rs12566098, and rs6702742 *SERBP1* in study groups are presented in Appendix A. The distribution of genotype frequencies of all studied SNPs corresponded to the Hardy–Weinberg equilibrium in both the control and case groups (*p* > 0.05).

The analysis of the total sample revealed a significant association between SNP rs12566098 *SERBP1* and IS risk (OR = 1.28, 95% CI 1.10–1.49, *p* = 0.001) (Table 1).

Owing to a potential sex difference in the associations of genetic variations with the risk of IS, the relationship between *SERBP1* polymorphisms and the emergence of IS was examined separately in males and females. The genetic variant rs12566098 was found to affect the risk of the disease in both males (OR = 1.25; 95% CI = 1.02–1.53; *p* = 0.03) and females (OR = 1.39; 95% CI = 1.14–1.71; *p* = 0.002) (Table 2). Meanwhile, rs1058074 was associated with the increased risk of IS exclusively in females (OR = 1.29, 95% CI 1.05–1.58, *p* = 0.02) (Table 2).

The analysis of relationships between *SERBP1* SNPs and the risk of IS depending on the smoking status revealed that rs1058074 (OR = 1.32, 95%CI 1.10–1.59, *p* = 0.003) and rs12566098 (OR = 1.43, 95%CI 1.20–1.72, *p* = 1 × 10^−4^) were associated with an increased risk of IS only in non-smokers (Table 2). Since revealed association between the genetic polymorphism and smoking-status may be biased by the disequilibrial proportion of smokers between males and females, we further performed separated analysis accounting both factors. The analysis has confirmed that smoking status has its own impact because we found that the association of rs1058074 and rs12566098 with the higher risk of IS in non-smokers still took place even after the exclusion of males (Appendix A). Moreover, we found that two other *SERBP1* SNPs, rs12561767 and rs6702742, were associated with a higher risk of IS in non-smoking men (OR = 1.36, 95% CI 1.05–1.77, *p* = 0.02, and OR = 1.33, 95% CI 1.03–1.73, *p* = 0.03, respectively) (Appendix A).

The influence of other environmental factors that may have a substantial impact on the development of IS, such as physical activity and dietary intake of fresh fruits and vegetables, was also taken into account for evaluating associations. Yet, independent of the effect of these factors, rs12566098 *SERBP1* was found to be related with the development of IS. (Table 2). Meanwhile, rs1058074 *SERBP1* showed associations with an increased risk of IS in patients who indicated a low level of physical activity before cerebrovascular events (OR = 1.26, 95% CI 1.04–1.54, *p* = 0.02, Pbonf = 0.04) (Table 2) and in patients who noted insufficient consumption of fresh vegetables and fruits (OR = 1.24, 95% CI 1.04–1.48, *p* = 0.02, Pbonf = 0.04) (Table 2), suggesting that these environmental risk factors, along with smoking, modify the association of rs1058074 with IS risk.

Moreover, we examined how the body mass index (BMI) of IS patients affected the link between the researched SNPs and the emergence of the condition. It was found that the carriage of the minor allele C rs1058074 was associated with a 1.5-fold increase in the risk of IS in a patient with a BMI ≥ 25 (OR = 1.47, 95% CI 1.14–1.89, *p* = 0.0045, Pbonf = 0.007). Similar data were obtained in the analysis of rs12566098: OR = 1.52, 95% CI 1.19–1.95, *p* = 0.001, Pbonf = 0.002 (Table 2).

### 2.2. Associations of SERBP1 with the Clinical Features of IS

All of the tested SNPs showed associations with the Activated Partial Thromboplastin Time (APTT). Interestingly, homozygotes for major alleles (protective against IS) had the highest APTT rates, while heterozygotes were characterized by intermediate APTT indicators. Homozygotes for minor alleles (risk alleles) had the lowest APTT values, suggesting that coagulation has been stimulated and that these patients had a tendency towards thrombosis. It is significant that consumption of fresh vegetables and fruits affects the relationship between rs1058074, rs12561767, rs12566098, and rs6702742 and the APTT (Appendix A, Figure 1).

An analysis of the influence of the studied *SERBP1* genotypes on body mass index established a relationship between rs4655707, rs12561767, and rs6702742 with this indicator only in patients with normal BMI, suggesting that *SERBP1* genetic variants lose their regulatory effects on BMI in patients with overweight. It is noteworthy that the association of rs4655707 with BMI was also observed in the group of patients with normal fruit and vegetable intake (Appendix A, Figure 1).

### 2.3. Bioinformatics Analysis

GTEx Portal claims that the *SERBP1* gene is expressed in brain tissues, blood vessels, and whole blood. In brain tissues, *SERBP1* expression levels measured as median TPM (Transcripts Per Million) vary from 18.16 to 43.22; in blood vessels, from 70.52 to 246.5; and in whole blood, MeTPM = 79.88 (https://gtexportal.org/home/gene/SERBP1 (accessed on 21 January 2023)).

#### 2.3.1. QTL-Effects

The results of the eQTL analysis for the studied *SERBP1* SNPs are shown in Table 3. According to the GTEx Portal browser, all studied SNPs are associated with expression of the *SERBP1* gene in the arteries (with cis-eQTL-mediated up-regulating effects of risk alleles) (Table 3). In the meantime, *SERBP1* SNPs exhibit bidirectional effects on the cis-eQTL-mediated *IL12RB2* level; they have a positive correlation with *IL12RB2* expression in the arteries and aorta and a negative correlation with *IL12RB2* expression in whole blood (Table 3).

Similar data were obtained in the analysis of eQTL effects using the eQTLGene browser data resource, which presents the results of the analysis of cis-eQTL effects in the blood: risk alleles of *SERBP1* SNPs are associated with a cis-eQTL-mediated increase in *IL12RB2* expression in blood (Table 3). Thus, the minor alleles of the studied *SERBP1* SNPs (risk alleles) affect the increase in *SERBP1* expression in the artery-tibial, the decrease in the expression of *IL12RB2* in the artery-tibial, and the increase in the expression of *IL12RB2* in the blood.

Analysis of cis-mQTL effects established a cis-mQTL-mediated effect of the risk alleles C rs1058074 and C rs12566098 on the decrease in methylation of cg24364144 (chr1:68102479) in the brain prefrontal cortex. At the same time, protective alleles C rs4655707 and G rs6702742 influenced the increase in methylation of cg24364144 (chr1:68102479) in the brain prefrontal cortex (Table 4). Given the fact that a decrease in methylation leads to an increase in gene expression, it can be concluded that the risk alleles of all IS-linked *SERBP1* SNPs are associated with a decrease in cg24364144 (chr1:68102479) methylation and, accordingly, an increase in *SERBP1* expression in the brain prefrontal cortex.

#### 2.3.2. Analysis of Transcription Factors

The analysis of transcription factors revealed that the protective allele C rs4655707 *SERBP1* creates DNA-binding sites for 60 TFs, co-controlling positive regulation of cysteine-type endopeptidase activity involved in the apoptotic signaling pathway (GO:0006919), response to hypoxia (GO:0001666), and cellular response to growth factor stimulus (GO:0071363) (Appendix A). 

Protective allele T rs1058074 *SERBP1* creates DNA-binding sites for 127 TFs that are jointly involved in 24 overrepresented GO controlling neurogenesis, cardio- and vasculogenesis, apoptosis, and cell signaling: cerebral cortex GABAergic interneuron fate commitment (GO:0021892); neurogenesis (GO:0022008); regulation of cell death (GO:0010941); cardiac muscle tissue regeneration (GO:0061026); forebrain dorsal/ventral pattern formation (GO:0021798); BMP signaling pathway involved in heart development (GO:0061312); cellular response to growth factor stimulus (GO:0071363); forebrain neuron fate commitment (GO:0021877); hypothalamus development (GO:0021854); neuroblast differentiation (GO:0014016); forebrain morphogenesis (GO:0048853); cardiac pacemaker cell development (GO:0060926); negative regulation of oligodendrocyte differentiation (GO:0048715); negative regulation of neurogenesis (GO:0050768); ventricular cardiac muscle cell development (GO:0055015); cardiac atrium morphogenesis (GO:0003209); neuron fate specification (GO:0048665); oligodendrocyte differentiation (GO:0048709); neural precursor cell proliferation (GO:0061351); heart valve development (GO:0003170); neuron migration (GO:0001764); hippocampus development (GO:0021766); regulation of neural precursor cell proliferation (GO:2000177); and blood vessel morphogenesis (GO:0048514) (Appendix A). Meanwhile, the risk allele C rs1058074 *SERBP1* creates DNA-binding sites for 18 TFs jointly involved in the regulation of protein stability (GO:0031647), neurogenesis processes such as forebrain neuron development (GO:0021884), neuron fate commitment (GO:0048663), neuron migration (GO:0001764) (Appendix A). 

Protective allele A rs12561767 *SERBP1* creates DNA-binding sites for 38 TFs characterized by the following overrepresented GO involved in cell signaling, apoptosis, and neurogenesis: steroid hormone mediated signaling pathway (GO:0043401); apoptotic process (GO:0006915); programmed cell death (GO:0012501); cell death (GO:0008219); neuron migration (GO:0001764); nervous system development (GO:0007399); and the generation of neurons (GO:0048699) (Appendix A). Risk allele G rs12561767 *SERBP1* creates DNA-binding sites for 15 TFs involved in the processes of programmed cell death: regulation of the apoptotic process (GO:0042981); and regulation of programmed cell death (GO:0043067) (Appendix A).

Risk allele C rs12566098 *SERBP1* creates DNA-binding sites for 62 TFs, co-controlling cellular response to growth factor stimulus (GO:0071363) and positive regulation of cytokine production (GO:0001819). At once, no IS-specific GO was found for nine TFs, binding to the protective allele G rs12566098 *SERBP1* (Appendix A).

Risk allele A rs6702742 *SERBP1* creates DNA-binding sites for 41 TFs that jointly involved in the positive regulation of neuron apoptotic process (GO:0043525), positive regulation of cytokine production (GO:0001819), and negative regulation of apoptotic process (GO:0043066). Meanwhile, protective allele G rs6702742 *SERBP1* creates DNA-binding sites for 23 TFs characterized by the following overrepresented GO: positive regulation of neuron death (GO:1901216); and regulation of neurogenesis (GO:0050767) (Appendix A).

#### 2.3.3. Bioinformatic Analysis of the Associations of SERBP1 SNPs with Cerebral Stroke and Intermediate Phenotypes

According to the bioinformatic resources, the Cerebrovascular Disease Knowledge Portal (CDKP) and the Cardiovascular Disease Knowledge Portal (CVDKP), which combine and analyze the results of genetic associations of the largest consortiums for the study of cardio- and cerebrovascular diseases, the IS-related *SERBP1* SNPs are associated with stroke and a number of stroke-related intermediate phenotypes (Table 5).

#### 2.3.4. Protein–Protein Interactions

Using the STRING database, we identified 10 proteins characterized by the most pronounced interactions with SERBP1 (proteins of the first shell of interactors): PGRMC1, RPL21, RPL23, RPL5, RPL8, RPS19, RPS24, RPS25, RPS29, and RPS6 (PPI enrichment *p*-value = 1.54 × 10^−10^) (Figure 2). These proteins are characterized predominantly by co-expression. Moreover, the interactions between these genes were established experimentally and using the textmining approaches. It is noteworthy that most of the main interaction partners of SERBP1 (RPS29, RPS24, RPL21, RPL5, RPS6, RPL23, RPS25, RPL8, RPS19) are directly involved in protein synthesis since they are structural components of ribosomes (Appendix A). The role of the PGRMC1 protein in the most significant interactions with SERBP1 should also be noted, since it is involved in the metabolism of progesterone and may play an essential role in the manifestation of the sex-specific effects of *SERBP1* (Appendix A).

Together, SERBP1 and its main interaction partners are involved in 14 biological processes and 6 molecular functions, reflecting not only the regulation of protein synthesis but also cotranslation, transport, ubiquitination, regulation of gene expression, the nitrogen compound biosynthetic process, and organic cyclic compound metabolism (Appendix A).

## 3. Discussion

The present study shows for the first time that polymorphisms in the gene encoding SERPINE1 mRNA-binding protein 1 (*SERBP1*) are associated with the risk and clinical features of IS. In particular, tag SNP rs12566098 (risk allele C) was found to be associated with an elevated risk of IS in the total study population, in the group of males and in the group of females, regardless of the level of physical activity (both in patients with low physical activity and in patients with a normal level of physical activity). Nonetheless, we noticed that cigarette smoking and consumption of fresh vegetables and fruits act as modifiers of the associations of rs12566098 with IS: this genetic variant is associated with IS only in non-smokers and individuals with low fruit and vegetable consumption. In addition, the analysis based on the BMI revealed that rs12566098 correlates with IS among overweight individuals (BMI ≥ 25). Most likely, this genetic variant imparts more risk as compared to other SNPs linked to it; their risk effects were modified by all the studied environmental factors, sex and the BMI. In particular, rs1058074 (risk allele C) is associated with an increased risk of IS exclusively in women, non-smokers, individuals with low physical activity, individuals with low fruit and vegetable consumption, and individuals with the BMI ≥ 25. SNPs rs12561767 (risk allele G) and rs6702742 (risk allele A) proved to be associated with an elevated risk of IS only among non-smoking males.

### 3.1. SERBP1 SNPs and the IS Risk: Underlying Mechanisms

*SERBP1* is expressed in vessels, brain tissues, and blood. *SERBP1* regulates mRNA degradation and formation of RNA–protein complexes, affects mRNA stability, and controls the mRNA levels of target genes [16]. According to GTex Portal data, risk alleles of all IS-associated SNPs (rs12561767, rs12566098, rs6702742, and rs1058074) are linked to cis-eQTL-mediated up-regulation of *SERBP1*. Further evidence of associations between risk alleles and *SERBP1* overexpression comes from the finding that risk alleles of all IS-related SNPs are associated with mQTLs, thus contributing to low methylation of CpG site cg24364144 (chr1:68102479) in the prefrontal cortex.

Because it binds to an AU-rich element in the 3′-untranslated region of PAI-1 mRNA and controls its stability, *SERBP1* causes PAI-1 protein levels to increase. Accordingly, it has been previously known as PAI-1 mRNA-binding protein 1 [16]. Plasminogen activator inhibitor 1 (PAI-1), which is a SERPIN inhibitor, is well known as a regulator of fibrinolysis. A high serum level of PAI-1 correlates with enhanced thrombosis by reducing fibrin degradation [23] and is seen in stroke patients [24]. Moreover, PAI-1 is involved in mechanisms of atherosclerotic plaque formation [25] and stabilization [26]. J.H. Heaton and co-authors have proposed that nuclear SERBP1 stabilizes PAI-1 mRNA and that cytoplasmic SERBP1 destabilizes it [27]. Most likely, the carriage of risk alleles associated with higher *SERBP1* expression contributes to the up-regulation of PAI-1. This notion is partially confirmed by our finding of the lowest APTT values in homozygotes for risk alleles of all the studied SNPs. It has been previously shown that hypofibrinolytic conditions are associated with shorter APTT and greater PAI-1 expression [28,29] and that PAI-1 overexpression correlates with a decrease in APTT [30]. Thus, our study indicates that risk alleles of *SERBP1* SNPs may be risk factors for excessive thrombosis. Nonetheless, their role in the PAI-1–mediated mechanisms of thrombosis in IS remains incompletely understood, and to assess the influence of *SERBP1* genotypes on the level of PAI-1 expression, functional studies are required: an analysis of the magnitude of PAI-1 expression depending on *SERBP1* genotypes. 

Because, to date, there have been no studies evaluating an association of *SERBP1* SNPs with the IS risk, a bioinformatic analysis was carried out here to interpret the functional effects of genetic variants. First, it should be noted that protective alleles of SNPs that correlated with IS in our study also showed an association with a reduced risk of lacunar stroke (rs1058074), small artery occlusion (rs12566098), and cardio-aortic embolism (rs6702742), according to the Cerebrovascular Disease Knowledge Portal, which provides comprehensive genetic data on large numbers of patients with stroke from across the world (https://cd.hugeamp.org/ (accessed on 24 January 2023)). Second, data obtained using the bioinformatic resource, the Cardiovascular Disease Knowledge Portal (https://cvd.hugeamp.org/ (accessed on 24 January 2023)), showed that protective alleles of IS-associated SNPs reduce LDL cholesterol (rs1058074, rs12561767, rs12566098, and rs6702742), non-HDL cholesterol (rs12566098), and total cholesterol (rs1058074) levels and increase HDL3 cholesterol levels, thus pointing to the potentially significant participation of *SERBP1* polymorphic variants in the development of atherosclerosis and thrombosis. Third, all IS-associated SNPs seem to contribute to the expression of *IL12RB2* in the blood and arteries through cis-eQTL effects. IL12RB2 is involved in IL-35 control, which is important for atherosclerosis and inflammation [31] and has been shown to inhibit ischemia/hypoxia-induced angiogenesis, suggesting that this anti-inflammatory cytokine plays new roles at the recovery stage of angiogenesis [32]. 

Fourth, IS-associated SNPs affect transcriptional modulation by TFs and exert pronounced actions such as a loss or gain of function. For instance, carriage of the risk allele C rs1058074 leads to loss of DNA binding to the TFs that are predominantly involved in neurogenesis, cardio- and vasculogenesis, apoptosis, and cell signaling and creates binding sites for the TFs that jointly regulate “neuron migration”, “neuron fate commitment” and “regulation of protein stability”. The risk allele G rs12561767 causes a loss of binding to the TFs involved in the steroid hormone signaling pathway and neurogenesis (such as “neuron migration” and “formation of neurons”) and creates binding sites for the TFs driving processes of programmed cell death. Risk allele C rs12566098 *SERBP1* creates binding sites for the TFs, which co-control the “cellular response to a growth factor stimulus” and “positive regulation of cytokine production”. The rs6702742 *SERBP1* risk allele A generates binding sites for TFs that promote neuronal apoptosis and cytokine production. Allele C of SNP rs4655707 provides binding sites for TFs taking part in positive regulation of “cysteine-type endopeptidase activity involved in apoptotic signaling pathways”, “response to hypoxia”, and “cellular response to growth factor stimulus”. The above-mentioned biological processes may be implicated both in the risk of IS and in processes of ischemia-reperfusion.

### 3.2. Sex-Specific Correlates of SERBP1

Our study revealed that rs1058074 is associated with IS only in females. Other research has already shown that genetic markers have well-pronounced sexual dimorphism in their correlations with stroke [33,34,35,36]. First, sex hormones progesterone and 17β-estradiol both regulate *SERBP1* mRNA levels [37]. On the other hand, according to the STRING database, one of the most important interaction partners of SERBP1 is PGRMC1 (progesterone receptor membrane component 1), which mediates the anti-apoptotic action of progesterone and whose mRNA is a target of SERBP1 [38]. Thus, *SERBP1* is both involved in the metabolism of female sex hormones and subject to regulation by these hormones, which may mediate sex-specific effects in the association of rs1058074 (in *SERBP1*) with the IS risk.

### 3.3. Smoking-Associated Correlates of SERBP1

Our study indicates that rs1058074 and rs12566098 (in *SERBP1*) are associated with an increased risk of IS only among non-smokers. On the one hand, smoking is a substantial environmental risk factor for the IS, and may contribute more significantly to the disease’s onset than *SERBP1* genetic variants. As a result, the link between SNPs and the IS risk is clearly shown in non-smokers. On the other hand, smoking can have a major impact on *SERBP1* expression. By means of the bioinformatics resource, the Comparative Toxigenomic Database, it has been found that tobacco smoke pollution affects the expression of the SERBP1 protein in humans [39]. Nevertheless, the direction of this influence remains unclear. To further understand how SERBP1 interacts with various components of cigarette smoke, more research on *SERBP1* expression levels in response to smoking status is required.

### 3.4. Low-Physical-Activity–Associated Correlates of SERBP1

We observed that rs1058074 is associated with an increased risk of IS only among individuals with low physical activity. The latter is a known risk factor for IS and may affect inflammatory, thrombotic, and oxidative markers [40], whose levels are modulated by chaperones [41]. Evidence for this is an observation that aerobic exercise training decreases plasma SERPINE1 levels in humans [42]. This lifestyle intervention may help to reduce the pathological effects of *SERBP1* risk alleles potentially associated with elevated SERPINE1 (PAI-1) levels. Moreover, the level of SERPINE1, which is regulated by SERBP1, can be modulated by the magnitude of physical activity [43].

### 3.5. Low Fruit/Vegetable Intake–Related Correlates of SERBP1

The association of SNP rs1058074 with IS was found to be modified by a low intake of fresh vegetables and fruits. Nonetheless, the most important finding is that four SNPs showed an association with APTT in the presence of a risk factor such as low consumption of fresh vegetables and fruits, suggesting that the prothrombotic effects of *SERBP1* risk alleles are significantly modified by this risk factor. Low consumption of fresh vegetables and fruits primarily correlates with oxidative stress [44,45], the role of which in the risk of IS and IS-related phenotypes has been demonstrated in a substantial number of studies [6,46,47,48,49,50]. Plant-based diets usually contain phenolic acids, flavonoids, and carotenoids, which have strong antioxidant properties and therefore eliminate an excess of active oxygen from the body and protect cells from damage, thereby reducing the risk of cardiovascular diseases [51]. It should be pointed out that the vasculo-protective influence of polyphenols (the main contributors to the antioxidant capacity of fruits and vegetables) may be linked to their antithrombotic action [52]. Antithrombotic mechanisms may be related to inhibition of platelet aggregation [53] and lower plasma homocysteine levels [54]. We believe that the manifestation of the risk effects of *SERBP1* during low consumption of vegetables and fruits is primarily due to oxidative-stress–induced thrombosis, which is enhanced by the pro-thrombotic effects of risk alleles of *SERBP1*.

### 3.6. SERBP1 SNPs and the BMI

We noticed that rs1058074 and rs12566098 *SERBP1* were associated with higher IS risk exclusively in individuals with a BMI of 25 or more. By contrast, no such correlations were found among individuals with a normal BMI. Moreover, *SERBP1* SNPs were associated with a higher BMI among patients with a BMI < 25. This relation can be explained in terms of connections between obesity, leptin and the level of *SERBP1* expression.

Leptin is a relevant adipokine that is involved in the regulation of food intake [55] and manifests its physiological actions by inhibiting food intake via actions on certain receptors in the brain [56]. In particular, considerable evidence implicates leptin in the conveying of satiety signals to the feeding center [57,58]. A bioinformatic analysis showed that the risk alleles of all the IS-linked SNPs contribute to lower leptin levels and hence weaker inhibitory effects of leptin on appetite, thereby increasing body weight. Nonetheless, these regulatory effects may only be seen in patients of normal body weight. In overweight individuals, leptin resistance (due to mutations in the genes encoding leptin or its receptors [59]) represents an inability of leptin to exert its anorexigenic actions [60]. Naturally, *SERBP1*-mediated regulation of the BMI was observed in a patient with a normal BMI who does not have leptin resistance. 

In addition, obesity promotes this propensity for thrombosis. PAI-1 levels, regulated by *SERBP1*, positively correlate with obesity [61] and are significantly reduced by weight loss in obese individuals [62].

## 4. Materials and Methods

The outline of the study design is shown in Figure 3.

The study was carried out on an ethnically homogeneous sample of unrelated residents of Central Russia (mainly natives of the Kursk region) of Russian nationality, with a total number of 2060 (869 patients with IS and 1191 healthy individuals). The Ethical Review Committee of Kursk State Medical University approved the study protocol. All the participants gave written informed consent before their enrollment in this study, subject to the following inclusion criteria: self-declared Russian descent, with a birthplace inside of Central Russia [63].

Baseline and clinical characteristics of the study population are listed in Table 6.

The patients were enrolled in the study in two periods: at the Regional Vascular Center of Kursk Regional Clinical Hospital between 2015 and 2017 and at the Neurology Clinics of Kursk Emergency Medicine Hospital between 2010 and 2012 [64]. All the patients were examined by qualified neurologists. The results of the brain computed tomography and/or magnetic resonance imaging were used to make the diagnosis of IS during the acute phase of stroke. The patients were recruited consecutively. The IS patients were enrolled under the following exclusion criteria: hepatic or renal failure; endocrine, autoimmune, oncological, or other diseases that can cause an acute cerebrovascular event; intracerebral hemorrhage; hemodynamic or dissection-related stroke; and traumatic brain injury. All the patients with IS had a history of hypertension and received antihypertensive therapy.

According to the WHO recommendations, low fruit and vegetable consumption was defined as an intake of less than 400 g per day (excluding potatoes and other starchy tubers) [65]. Insufficient physical activity was defined as less than 180 min per week of physical activities of moderate to vigorous intensity. We included both physical activities performed in leisure time (for example, walking and running) and fitness club activities (for example, running on a treadmill, aerobics, or resistance training). It is this level of physical activity that is considered protective against the development of cardiovascular diseases by most authors [66,67,68]. According to the WHO criteria, overweight is defined as patients with a body mass index of 25 or over [69].

Healthy volunteers who had normal blood pressure without receiving antihypertensive treatment and no clinical signs of cardiovascular, cerebrovascular, or other serious illnesses comprised the control group. A healthy person met the criteria for inclusion in the control group if their systolic blood pressure was less than 130 mm Hg and their diastolic blood pressure was less than 85 mm Hg on at least three separate measurements. Control subjects were enrolled from Kursk hospitals during periodic medical examinations at public institutions and industrial enterprises of Kursk region [70,71]. This group was recruited from the same population and during the same period.

The selection of SNPs was based on the following criteria: the SNP must have a minor allele frequency of at least 0.05 in the European population and be characterized by a high regulatory potential. According to the bioinformatic tools SNPinfo Web Server (https://snpinfo.niehs.nih.gov/ (accessed on 19 March 2022)) and LD TAG SNP Selection (TagSNP), which were used to select SNPs based on the reference haplotypic structure of the Caucasian population (CEU) of the project HapMap, the *SERBP1* (SERPINE1 MRNA Binding Protein 1, ID:26135) gene contains five SNPs (rs4655707, rs1058074, rs12561767, rs12566098, rs6702742). SNPs rs4655707 and rs1058074 are located in three prime UTR; rs12561767, rs12566098, and rs6702742 are located in introns.

The regulatory potential of these SNPs was evaluated using a variety of bioinformatic tools. RegulomeDB tool showed that rs4655707, rs1058074, and rs6702742 SERBP1 are characterized by a regulatory coefficient of 5 (TF binding or DNase peak); rs12566098, by a regulatory coefficient of 4 (TF binding + DNase peak); rs12561767, by a regulatory coefficient of 1f (eQTL + TF binding / DNase peak) (http://regulome.stanford.edu/ (accessed on 20 March 2022)) [72]. HaploReg (v4.1) database showed that this SNPs are characterized by enhancer histone marks in different tissues (rs4655707, rs12561767, rs12566098, rs6702742), regions of hypersensitivity for DNAse 1 (rs12561767, rs12566098, rs6702742), binding site for regulatory proteins (rs4655707, rs12561767), and DNA regulatory motifs (rs1058074, rs12561767, rs6702742) (http://archive.broadinstitute.org/mammals/haploreg/haploreg.php (accessed on March 20, 2022)) [73].

According to the data presented by the NCBI source (https://www.ncbi.nlm.nih.gov/snp/ (accessed on 20 January 2023)), these genetic variants are characterized by an average frequency of the minor alleles in European populations of 0.30 (rs12566098) to 0.46 (rs4655707) (https://www.ensembl.org/ (on 20 March 2022)). Thus, all five SNPs were selected for our research, which meets the necessary criteria for inclusion in the study.

### 4.1. Genetic Analysis 

DNA analysis was carried out at the Laboratory of Genomic Research of Research Institute for Genetic and Molecular Epidemiology of Kursk State Medical University (Kursk, Russia). Approximately 5 mL of venous blood from the cubital vein of each participant was collected into EDTA-coated tubes and maintained at −20 ◦C until processed. Genomic DNA was extracted from thawed blood samples by the standard procedure of phenol/chloroform extraction and ethanol precipitation. A NanoDrop spectrophotometer (Thermo Fisher Scientific, Waltham, MA, USA) was used to evaluate the extracted DNA solution’s purity, quality, and concentration. 

Genotyping of the SNPs was conducted using allele-specific probe-based real-time polymerase chain reaction assays according to the protocols designed in the Laboratory of Genomic Research, Research Institute for Genetic and Molecular Epidemiology. Primers and probes were designed using the Primer3 program online (http://primer3.ut.ee/ (accessed on 30 March 2022)) [74], selected, and then synthesized by the Syntol company (Moscow, Russia). 

The primers and probes used for genotyping the polymorphisms are presented in Table S. 11. A real-time PCR was conducted in a 25-mL reaction mixture containing 1.5 U of Hot Start Taq DNA polymerase (Biolabmix, Novosibirsk, Russia), about 1 μg of DNA, 0.25 μM each primer, 250 μM dNTPs, 3.0 mM MgCl_2_ (for rs4655707), 2.5 mM MgCl_2_ (for rs1058074, rs12566098, rs6702742), 1.5 mM MgCl_2_ (for rs12561767), and 1xPCR buffer (67 mM Tris-HCl, pH 8.8, 16.6 mM (NH_4_)2SO_4_, 0.01% Tween-20). The amplification reaction consisted of an initial denaturation for 10 min at 95 °C, followed by 39 cycles of 92 °C for 30 s and 64 °C, 49 °C, 64 °C, 56 °C, 53 °C for 1 min (for rs4655707, rs1058074, rs12561767, rs12566098, and rs6702742, respectively). To ensure quality control, 10% of DNA samples were genotyped in duplicates, blinded to the case-control status. The concordance rate was >99%.

### 4.2. Statistical and Bioinformatics Analysis

The genetic association study power calculator, accessible online at http://csg.sph.umich.edu/abecasis/gas_power_calculator/ (accessed on 15 January 2023), was used to calculate the statistical power for the study. Association analysis between the *SERBP1* gene polymorphisms and IS risk could detect the genotype relative risk of 1.19–1.45 assuming 0.80 power and a 5% type-I error (α = 0.05) on the sample size of 869 cases and 1191 controls.

All statistical analyses were performed using the STATISTICA software (v13.3, USA). The distribution of quantitative data was tested for normality using Shapiro–Wilk’s test. Since the biochemical parameters and BMI showed a deviation from normal distribution, they were expressed as the median (Me) and first and third quartiles (Q1 and Q3). The Kruskal–Wallis test was used to compare quantitative variables among three independent groups. Following that, groups were contrasted pairwise using the Mann–Whitney test with FDR adjustment [75]. For categorical variables, the statistical significance of differences was evaluated by Pearson’s chi-squared test with Yates’s correction for continuity.

Compliance of the genotypes’ distributions with Hardy–Weinberg equilibrium was assessed using Fisher’s exact test. Genotype frequencies in the study groups and their associations with the disease risk were analyzed using SNPStats software (https://www.snpstats.net/start.htm (accessed on 15 January 2023)) [76]. For the analysis of associations among genotypes, additive models were considered. Associations in the entire group of IS patients/controls were adjusted for age, gender, and smoking status. In cases where there was no information about the environmental risk factor in the control group, associations were analyzed depending on the presence or absence of the risk factor in the group of patients, compared with the total control group. In this case, the Bonferroni correction was additionally introduced [77].

The following bioinformatics resources were used to analyze the functional effects of *SERBP1* SNPs:The bioinformatic tool GTExportal (http://www.gtexportal.org/ (accessed on 21 January 2023)) was used to analyze the expression levels of the studied genes in the brain, whole blood, and blood vessels, as well as to analyze expression quantitative trait loci (eQTLs) (The GTEx Consortium, 2020). Additionally, the eQTLGene browser (https://www.eqtlgen.org/ (accessed on 21 January 2023)) was used to analyze the cis-eQTL-mediated effects of *SERBP1* SNPs in blood [78];The methylation quantitative trait loci (mQTLs) in the brain, whole blood, and blood vessels were examined using QTLbase (http://www.mulinlab.org/qtlbase/index.html (accessed on 21 January 2023)) [79];Bioinformatic tools of the STRING database (https://string-db.org/ (accessed on 22 January 2023)) were used for the analysis of the main interaction partners of SERBP1 [80]. Analysis of biological processes and molecular functions reflecting interactions with main functionally related proteins also was carried out in STRING database;The atSNP Function Prediction online tool (http://atsnp.biostat.wisc.edu/search (accessed on 22 January 2023)) was used to evaluate the impact of *SERBP1* SNPs on the binding of transcription factors (TFs) to DNA depending on the carriage of the reference/alternative alleles [81]. TFs were included based on the degree of influence of SNPs on the interaction of TFs with DNA, calculated on the basis of a positional weight matrix;Using the Gene Ontology online tool (http://geneontology.org/ (accessed on 23 January 2023)), it was feasible to analyze the joint involvement of TFs linked to the reference/SNP alleles in overrepresented biological processes directly related to the pathogenesis of IS [82]. Biological functions controlled by transcription factors associated with *SERBP1* SNPs were used as functional groups;The Comparative Toxicogenomics Database (CTD) resource (http://ctdbase.org (accessed on 24 January 2023)) was used for the interpretation of environment-associated correlates of *SERBP1*. CTD provides the ability to analyze specific interactions between genes and chemicals in vertebrates and invertebrates based on data obtained from published scientific studies worldwide [83]. This tool was used to analyze binary interactions involving one chemical and one gene or protein;The Cerebrovascular Disease Knowledge Portal (CDKP) (https://cd.hugeamp.org/ (accessed on 24 January 2023)) and Cardiovascular Disease Knowledge Portal (https://cvd.hugeamp.org/ (accessed on 24 January 2023)) online tools were used for bioinformatic analyses of the associations of *SERBP1* SNPs with atherosclerosis-associated diseases, intermediate phenotypes, and risk factors for IS (such as total cholesterol, LDL, BMI, etc.).

## Figures and Tables

**Figure 1 ijms-24-08716-f001:**
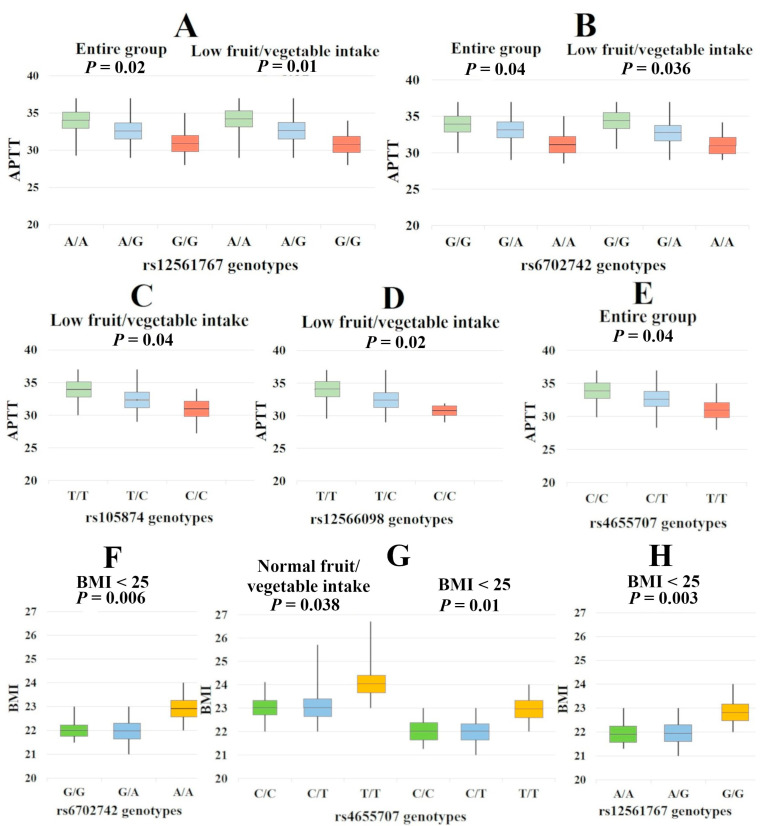
Associations of *SERBP1* genotypes with activated partial thromboplastin time and body mass index. (**A**)—APTT values for rs12561767 in the entire group (*p* = 0.02) and in the group of patients with low fruit/vegetable intake (*p* = 0.01); (**B**)—APTT values for rs6702742 in the entire group (*p* = 0.04) and in the group of patients with low fruit/vegetable intake (*p* = 0.036); (**C**)—APTT values for rs105874 in the group of patients with low fruit/vegetable intake (*p* = 0.04); (**D**)—APTT values for rs12566098 in the group of patients with low fruit/vegetable intake (*p* = 0.02); (**E**)—APTT values for rs4655707 in the entire group (*p* = 0.04); (**F**)—BMI values for rs6702742 in the group of patients with BMI < 25 (*p* = 0.006); (**G**)—BMI values for rs4655707 in the group of patients with normal fruit/vegetable intake (*p* = 0.038) and in the group of patients with BMI < 25 (*p* = 0.01); (**H**)—BMI values for rs12561767 in the group of patients with BMI < 25 (*p* = 0.0023).

**Figure 2 ijms-24-08716-f002:**
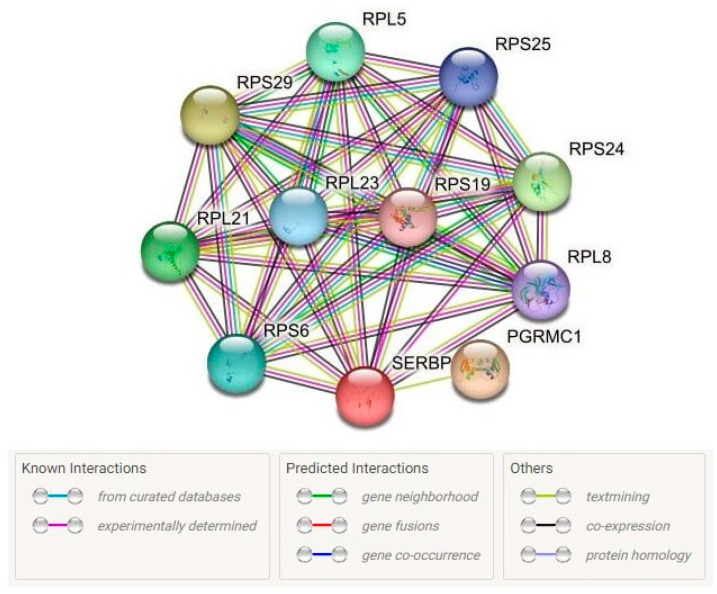
Predicted Interaction Partners of SERBP1 (https://string-db.org/ (accessed on 22 January 2023)).

**Figure 3 ijms-24-08716-f003:**
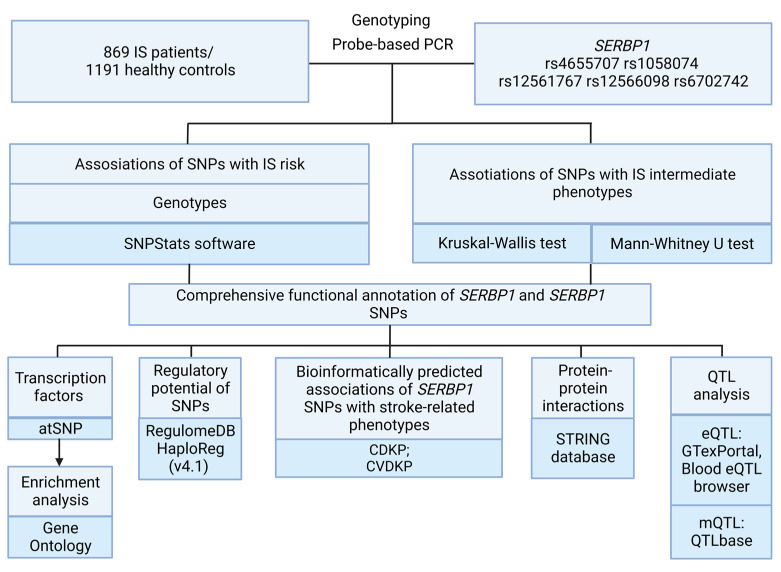
Study design.

**Table 1 ijms-24-08716-t001:** Results of the analysis of associations between *SERBP1* gene SNPs and ischemic stroke risk.

Genetic Variant	Effect Allele	Other Allele	N	OR (95% CI) ^1^	*p* ^2^
rs4655707	T	C	1883	1.07 [0.93; 1.23]	0.35
rs1058074	C	T	1915	1.15 [0.99; 1.35]	0.071
rs12561767 *	G	A	1857	1.11 [0.96; 1.27]	0.16
rs12566098 *	C	G	2060	**1.28 [1.10; 1.49]**	**0.001**
rs6702742	A	G	1898	1.04 [0.91; 1.20]	0.54

All calculations were performed relative to the minor alleles (Effect allele) with adjustment for sex, age, smoking; ^1^—odds ratio and 95% confidence interval; ^2^—*p*-value; tag SNPs are marked with an asterisk; statistically significant differences are marked in bold.

**Table 2 ijms-24-08716-t002:** Subgroups analysis of associations between *SERBP1* SNPs and IS risk depending on sex, smoking status, body mass index, physical activity level, and fruit and vegetable intake.

Genetic Variant	Effect Allele	Other Allele	N	OR (95% CI) ^1^	*p* ^2^ (P_bonf_)	N	OR (95% CI) ^1^	*p* ^2^ (P_bonf_)
			Males			Females		
rs4655707	T	C	929	1.08 [0.90; 1.31]	0.39	954	1.07 [0.88; 1.29]	0.51
rs1058074	C	T	950	1.12 [0.91; 1.38]	0.29	965	**1.29 [1.05; 1.58]**	**0.02**
rs12561767 *	G	A	918	1.12 [0.93; 1.35]	0.23	939	1.09 [0.90; 1.32]	0.39
rs12566098 *	C	G	1026	**1.25 [1.02; 1.53]**	**0.03**	1124	**1.39 [1.14; 1.71]**	**0.002**
rs6702742	A	G	939	1.10 [0.92; 1.32]	0.29	959	1.00 [0.82; 1.20]	0.97
			Nonsmokers			Smokers		
rs4655707	T	C	1281	1.11 [0.94; 1.31]	0.22	602	0.93 [0.73; 1.20]	0.59
rs1058074	C	T	1317	**1.32 [1.10; 1.59]**	**0.003**	598	0.87 [0.67; 1.14]	0.32
rs12561767 *	G	A	1277	1.18 [1.00; 1.39]	0.054	580	0.94 [0.73; 1.21]	0.62
rs12566098 *	C	G	1400	**1.43 [1.20; 1.72]**	**1 × 10^−4^**	660	1.02 [0.79; 1.32]	0.89
rs6702742	A	G	1305	1.09 [0.93; 1.29]	0.29	593	0.92 [0.72; 0.72]	0.49
			Low physical activity (f+)			Normal physical activity level (f-)		
rs4655707	T	C	1387	1.07 [0.89; 1.28]	0.46 (0.92)	1546	1.06 [0.90; 1.24]	0.48 (0.96)
rs1058074	C	T	1400	**1.26 [1.04; 1.54]**	**0.02** (**0.04**)	1567	1.14 [0.96; 1.36]	0.14 (0.28)
rs12561767 *	G	A	1364	1.17 [0.98; 1.40]	0.086 (0.172)	1526	1.05 [0.90; 1.23]	0.54 (1.0)
rs12566098 *	C	G	1525	**1.39 [1.15; 1.68]**	**9 × 10^−4^** (**0.002**)	1692	**1.26 [1.06; 1.49]**	**0.009** (**0.02**)
rs6702742	A	G	1411	1.10 [0.93; 1.31]	0.3 (0.6)	1564	1.00 [0.85; 1.17]	0.99 (1.0)
			Low fruit/vegetable intake (f+)			Normal fruit/vegetable intake (f-)		
rs4655707	T	C	1492	1.08 [0.92; 1.28]	0.32 (0.64)	1441	1.04 [0.87; 1.23]	0.68 (1.0)
rs1058074	C	T	1514	**1.24 [1.04; 1.48]**	**0.02** (**0.04**)	1453	1.13 [0.93; 1.37]	0.21 (0.42)
rs12561767 *	G	A	1468	1.08 [0.92; 1.28]	0.33 (0.66)	1422	1.11 [0.94; 1.32]	0.22 (0.44)
rs12566098*	C	G	1638	**1.37 [1.15; 1.63]**	**4 × 10^−4^** (**8 × 10^−4^**)	1579	**1.24 [1.02; 1.49]**	**0.03** (0.06)
rs6702742	A	G	1510	1.04 [0.89; 1.22]	0.62 (1.0)	1465	1.04 [0.87; 1.23]	0.67 (1.0)
			BMI ≥ 25			BMI < 25		
rs4655707	T	C	1234	1.20 [0.95; 1.53]	0.13 (0.26)	1443	1.01 [0.85; 1.20]	0.91 (1.0)
rs1058074	C	T	1252	**1.47 [1.14; 1.89]**	**0.0045** (**0.007**)	1456	1.03 [0.85; 1.25]	0.74 (1.0)
rs12561767 *	G	A	1213	1.25 [0.98; 1.59]	0.071 (0.142)	1430	1.03 [0.87; 1.22]	0.71 (1.0)
rs12566098 *	C	G	1366	**1.52 [1.19; 1.95]**	**0.001** (**0.002**)	1579	1.17 [0.97; 1.41]	0.096 (0.192)
rs6702742	A	G	1272	1.22 [0.97; 1.54]	0.091 (0.182)	1478	1.00 [0.84; 1.18]	0.99 (1.0)

^1^—odds ratio and 95% confidence interval; ^2^—*p*-value (*p*-value with Bonferroni correction). All calculations were performed relative to the minor allele (Effect allele). Statistically significant differences are marked in bold; tag SNPs are marked with an asterisk.

**Table 3 ijms-24-08716-t003:** Relationship between *SERBP1* SNPs and cis-eQTL-mediated expression levels of genes in IS-related tissues (blood vessels, brain, whole blood) (according to browsers’ GTEx Portal, eQTLGene).

SNP	GTEx Portal Data (https://gtexportal.org (Accessed on 21 January 2023))					eQTLGene Browser Data (https://www.eqtlgen.org/ (Accessed on 21 January 2023))			
	Effect Allele	Gene Expressed	*p*-Value	Effect (NES)	Tissue	Effect Allele	Gene Expressed	Z-Score	*p*-Value
rs4655707 ***SERBP1*** (T/C)	C	*IL12RB2*	**6.2 × 10^−10^**	↑(0.28)	Artery—Tibial	T	*IL12RB2*	↑(25.93)	**2.93 × 10^−148^**
		*SERBP1*	**1.8 × 10^−6^**	↓(−0.091)	Artery—Tibial				
		*IL12RB2*	**3.8 × 10^−8^**	↓(−0.13)	Whole Blood				
rs1058074 ***SERBP1*** (C/T)	T	*IL12RB2*	**1.5 × 10^−11^**	↑(0.30)	Artery—Tibial	C	*IL12RB2*	↑(16.31)	**9.01 × 10^−60^**
		*SERBP1*	**8.7 × 10^−10^**	↓(−0.12)	Artery—Tibial				
		*IL12RB2*	**5.6 × 10^−7^**	↓(−0.12)	Whole Blood				
		*IL12RB2*	**3.6 × 10^−5^**	↑(0.25)	Artery—Aorta				
rs12561767 * ***SERBP1*** (G/A)	A	*IL12RB2*	**2.8 × 10^−10^**	↑(0.28)	Artery—Tibial	G	*IL12RB2*	↑(26.20)	**2.92 × 10^−151^**
		*IL12RB2*	**9.1 × 10^−9^**	↓(−0.14)	Whole Blood				
		*SERBP1*	**2.3 × 10^−6^**	↓(−0.09)	Artery—Tibial				
rs12566098 * ***SERBP1*** (C/G)	G	*SERBP1*	**1.3 × 10^−9^**	↓(−0.12)	Artery—Tibial	C	*IL12RB2*	↑(17.40)	**8.60 × 10^−68^**
		*IL12RB2*	**3.7 × 10^−7^**	↓(−0.13)	Whole Blood				
		*IL12RB2*	**2.3 × 10^−11^**	↑(0.30)	Artery—Tibial				
rs6702742 ***SERBP1*** (A/G)	G	*IL12RB2*	**3.9 × 10^−10^**	↑(0.27)	Artery—Tibial	A	*IL12RB2*	↑(26.29)	**2.42 × 10^−152^**
		*IL12RB2*	**1.0 × 10^−9^**	↓(−0.14)	Whole Blood				
		*SERBP1*	**9.5 × 10^−7^**	↓(−0.092)	Artery—Tibial				

tag SNPs are marked with an asterisk.

**Table 4 ijms-24-08716-t004:** Established associations of the studied *SERBP1* SNPs with the cis-mQTL-mediated effect on the level of methylation of CpG-sites.

Trait	Effect Allele	Tissue	Effect Size (Beta)	FDR
rs4655707 *SERBP1*				
cg24364144 (chr1:68102479)	C	Brain-Prefrontal Cortex	0.02	9.5 × 10^−14^
rs1058074 *SERBP1*				
cg24364144 (chr1:68102479)	C	Brain-Prefrontal Cortex	−0.01	1.6 × 10^−10^
rs12561767 *SERBP1* *				
cg24364144 (chr1:68102479)	A	Brain-Prefrontal Cortex	0.02	9.5 × 10^−14^
rs12566098 *SERBP1* *				
cg24364144 (chr1:68102479)	C	Brain-Prefrontal Cortex	−0.01	5.1 × 10^−11^
rs6702742 *SERBP1*				
cg24364144 (chr1:68102479)	G	Brain-Prefrontal Cortex	0.01	6.5 × 10^−13^

Note: mQTL with an undetermined effect allele was excluded from the analysis. tag SNPs are marked with an asterisk.

**Table 5 ijms-24-08716-t005:** Results of aggregated bioinformatic analyses of associations between SNPs in *SERBP1* gene cerebrovascular diseases and their intermediate phenotypes.

№	SNPs	Phenotype	*p*-Value	Beta (OR)	Sample Size
1.	rs4655707 (T/**C**)	^1^ TOAST other undetermined	0.01	_OR_▼ 0.7757	9487
2.		^1^ TOAST cardio-aortic embolism	0.02	_OR_▼ 0.7946	9470
3.		^1^ Lacunar stroke	0.043	_OR_▼ 0.9611	28,530
4.		^2^ LDL cholesterol	0.015	_Beta_▼ −0.0031	2,028,070
5.		^2^ Leptin	0.038	_Beta_▲ 0.0097	36,525
6.	rs1058074 (C/**T**)	^1^ Lacunar stroke	0.020	_OR_▼ 0.9546	28,530
7.		^1^ TOAST other undetermined	0.021	_OR_▼ 0.9546	9487
8.		^2^ Arm fat ratio	0.007	_Beta_▼ −0.0013	232,276
9.		^2^ Dyslipidemia	0.020	_OR_▲ 1.0322	56,375
10.		^2^ LDL cholesterol	0.020	_Beta_▼ −0.0029	2,010,200
11.		^2^ Leptin	0.037	_Beta_▲ 0.0108	32,169
12.		^2^ Non-HDL cholesterol	0.042	_Beta_▼ −0.0032	1,096,010
13.		^2^ Total cholesterol	0.049	_Beta_▼ −0.0026	1,921,240
14.	rs12561767 * (G/**A**)	^1^ TOAST other undetermined	0.010	_OR_▼ 0.7775	9487
15.		^2^ LDL cholesterol	0.010	_Beta_▼ −0.0032	2,014,850
16.		^2^ Dyslipidemia	0.040	_OR_▲ 1.0264	56,375
17.		^2^ Leptin	0.036	_Beta_▲ 0.0098	36,521
18.	rs12566098 * (C/**G**)	^1^ TOAST small artery occlusion	0.047	_OR_▲ 2.5943	254,558
19.		^2^ LDL cholesterol	0.010	_Beta_▼ −0.0032	2,021,740
20.		^2^ Arm fat ratio	0.010	_Beta_▼ −0.0112	232,276
21.		^2^ Dyslipidemia	0.020	_OR_▲ 1.0315	56,375
22.		^2^ Leptin	0.020	_Beta_▲ 0.0118	34,005
23.		^2^ Non-HDL cholesterol	0.030	_Beta_▼ −0.0035	1,108,320
24.	rs6702742 (A/**G**)	^1^ TOAST other undetermined	0.009	_OR_▼ 0.7726	9487
25.		^1^ TOAST cardio-aortic embolism	0.025	_OR_▼ 0.7952	9470
26.		^2^ Leptin	0.002	_Beta_▲ 0.0188	29,651
27.		^2^ HDL3 cholesterol	0.021	_Beta_▲ 0.0336	10,984
28.		^2^ LDL cholesterol	0.024	_Beta_▼ −0.0027	2,144,190
29.		^2^ Leptin adj BMI	0.028	_Beta_▲ 0.0170	22,924
30.		^2^ Dyslipidemia	0.044	_OR_▲ 1.0264	56,375

^1^—data obtained using the bioinformatic resource Cerebrovascular Disease Knowledge Portal (https://cd.hugeamp.org/ (accessed on 24 January 2023)); ^2^—data obtained using the bioinformatic resource Cardiovascular Disease Knowledge Portal (https://cvd.hugeamp.org/ (accessed on 24 January 2023)). Effect alleles are marked in bold; tag SNPs are marked with an asterisk.

**Table 6 ijms-24-08716-t006:** Baseline and clinical characteristics of the studied groups.

Baseline and Clinical Characteristics		IS Patients (*n* = 869)	Controls (*n* = 1191)	*p*-Value
Age, Me [Q1; Q3]		62 [55; 70]	59 [53; 66]	**<0.001**
Gender	Males, N (%)	482 (55.5%)	544 (45.7%)	**<0.001**
	Females, N (%)	387 (44.5%)	647 (54.3%)	
Smoking	Yes, N (%)	419 (48.2%)	241 (20.2%)	**<0.001**
	No, N (%)	450 (51.8%)	950 (79.8%)	
Low physical activity	Yes, N (%)	334 (40.0%)	ND	
	No, N (%)	501 (60.0%)		
Low fruit/vegetable consumption	Yes, N (%)	447 (53.53%)	ND	
	No, N (%)	388 (46.47%)		
Coronary artery disease	Yes, N (%)	264 (26.72%)	-	
	No, N (%)	724 (73.28%)	-	
Type 2 diabetes mellitus	Yes, N (%)	101 (12.11%)	-	
	No, N (%)	733 (87.89%)	-	
Body mass index, Me [Q1; Q3]		23 [22; 26] (*n* = 563)	-	
Overweight	Normal weight (BMI = 18.5–24.9), N (%)	388 (68.92%)	ND	
	Overweight (BMI of 25–29.9), N (%)	118 (20.96%)		
	Obesity (BMI of 30 or greater), N (%)	57 (10.12%)		
Family history of cerebrovascular diseases	Yes, N (%)	311 (34.44%)	ND	
	No, N (%)	592 (65.56%)	ND	
Age at onset of stroke, Me [Q1; Q3]		61 [53; 69] (*n* = 851)	-	
Number of strokes including event in question	1, N (%)	751 (88.35%)	-	
	2, N (%)	86 (10.12%)	-	
	3, N (%)	13 (1.53%)	-	
Type of stroke	Atherothrombotic, N (%)	616 (70.89%)	-	
	Cardioembolic, N (%)	160 (18.41%)	-	
	Unspecified, N (%)	93 (10.70%)	-	
Stroke localization	Right/left middle cerebral artery basin, N (%)	705 (83.04%)	-	
	Vertebrobasilar basin, N (%)	144 (16.96%)	-	
Area of lesion in stroke, mm^2^, Me [Q1; Q3]		99.50 [30; 461] (*n* = 776)	-	
Total cholesterol, mmol/L, Me [Q1; Q3]		5.2 [4.4; 5.8] (*n* = 577)	ND	
Triglycerides, mmol/L, Me [Q1; Q3]		1.3 [1.09; 1.80] (*n* = 571)	ND	
Glucose level, mmol/L, Me [Q1; Q3]		4.8 [4.3; 5.5] (*n* = 840)	ND	
Prothrombin time, seconds, Me [Q1; Q3]		10.79 [10.14; 11.70] (*n* = 827)	ND	
International normalized ratio, Me [Q1; Q3]		1 [0.93; 1.09] (*n* = 566)	ND	
Activated partial thromboplastin time, seconds, Me [Q1; Q3]		32.6 [29; 37] (*n* = 569)	ND	
Alanin aminotransferase, IU/L		21.9 [18; 31.2] (*n* = 646)	ND	
Aspartate aminotransferase, IU/L		28.2 [20.5; 37.4] (*n* = 646)	ND	

Statistically significant differences between groups are indicated in bold; ND—no data.

## Data Availability

The data presented in this study are available upon request from corresponding author.

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
