# Peer review of "SERPINE1 mRNA Binding Protein 1 Is Associated with Ischemic Stroke Risk: A Comprehensive Molecular–Genetic and Bioinformatics Analysis of SERBP1 SNPs"

_ijms, 2023, doi:10.3390/ijms24108716_

Round 1

Reviewer 1 Report

The manuscript looks to evaluate the role of SERPINE1 mRNA Binding Protein 1 (SERBP1) in the context of ischemic stroke. In particular, investigators examined DNA samples from 2060 unrelated Russian subjects and determined the role of various SERBP1 SNPs may have as novel genetic markers for ischemic stroke.

Overall, the manuscript is comprehensive and well put together. The data is well presented across the various tables and text. Some minor changes are recommended:

Figure 2: Provide additional context and add to figure legend. Example: n value or mouse vs human data etc.

Author Response

Respected editors and reviewers!

Let us express our sincere gratitude and deep thankfulness for your attention to our manuscript “SERPINE1 mRNA Binding Protein 1 is associated with ischemic stroke risk: a comprehensive molecular-genetic and bioinformatics analysis of SERBP1 SNPs”. Thank you for taking and spending your time on our article, for a thorough analysis of it and for the comments made.

Below we provide answers to the questions and comments of the reviewer:

Reviewer #1:

Figure 2: Provide additional context and add to figure legend. Example: n value or mouse vs human data etc.

Thank you very much for your comment. We withdrew the graphic displaying the SERBP1 expression data when one of the reviewers advised against doing so, leaving only textual information on the level of gene expression in its place.

Reviewer 2 Report

I congratulate you on this detailed and consistently conclusive study. Please keep in mind, however, that the number of objects on which the study is based is small in order to derive the generally valid statement to use the SERBP1 SNPs as a new marker in your paper without restriction. Please put this statement into perspective in the chapters "Abstract" and "Discussion".

Author Response

Respected editors and reviewers!

Let us express our sincere gratitude and deep thankfulness for your attention to our manuscript “SERPINE1 mRNA Binding Protein 1 is associated with ischemic stroke risk: a comprehensive molecular-genetic and bioinformatics analysis of SERBP1 SNPs”. Thank you for taking and spending your time on our article, for a thorough analysis of it and for the comments made.

Below we provide answers to the questions and comments of the reviewer:

Reviewer #2:

Please keep in mind, however, that the number of objects on which the study is based is small in order to derive the generally valid statement to use the SERBP1 SNPs as a new marker in your paper without restriction. Please put this statement into perspective in the chapters "Abstract" and "Discussion".

Thank you very much; we have added information about this limitation of our study to the abstract and "Limitations of the Study".

Reviewer 3 Report

The study by Shilenok et al. reports an association between SNPs in a regulator of serpine and stroke prevalence. The contribution of this study to the field is incremental but describes an under-studied population (Central Russia). Unfortunately, like in many similar bioinformatic studies, no functional contribution of the reported snips to stroke occurrence and progression can be implied. However, the manuscript is clearly written and does have any major methodological mistakes, only some over-interpretations. I, therefore, leave the decision about the suitability of this manuscript to the editorial team. My recommendations for improving the manuscript are the following:

1. Stroke types are not characterized. Hemorrhagic (subarachnoid and intracerebral), lacunar, hemodynamic, and ischemic strokes should be distinguished.

2. BMI analysis is too crude. Usually, obesity and not overweight is considered as a more prominent risk factor. I would recommend splitting 25-28, 28-30, and over 30. 

3. What is the meaning of vegetable intake? WHO recommendations are very general, and dietary parameters are highly regional. For Russia, may this be an indicator of income?

4. Statement about smoking lines 110-112 needs to be improved/explained. Smoking per se has a (weak) effect on stroke occurrence. So, probably smoking simply has a larger contribution than snps here.

5. Avoid statements that are not supported by data, for example,

lines 376-378: "Most likely, tobacco smoke reduces SERBP1 expression, thereby partially offsetting the risk-promoting effects of alleles associated with SERBP1overexpression". There is no evidence for that.

lines 426-427: "The association of rs1058074 and rs12566098 (in SERBP1) with IS exclusively among patients with a high BMI may be due to the influence of diet on SERBP1 expression". Referring to the experimental papers in mice afterward does not prove it. First, those were rodent studies, and second, analysis was mainly on the protein level

6. Figure 1: readability should be improved. Fonts, statistical differences, scales and color choices are not optimal

7. Figure 2:  I don't it's appropriate for publication, because this is barely a reproduction of a part of this figure https://gtexportal.org/home/gene/SERBP1

8. Adding a more comprehensive infographic on the geography, demographics, and clinical data of the patients/healthy persons is highly recommended

9. Terms and abbreviations:

line 167: explain MeTPM

line 269: Functional partners is misleading. Interaction partners, or gene network associations will be more appropriate.

10. Please distill the citation list: many are from less reliable journals and some are even not translated into English

Round 2

Reviewer 3 Report

Thanks for the revisions and good luck in the future studies